# Can Biomarkers Respond Upon Freshwater Pollution?—A Moss-Bag Approach

**DOI:** 10.3390/biology10010003

**Published:** 2020-12-22

**Authors:** Gana Gecheva, Ivelin Mollov, Galina Yahubyan, Mariyana Gozmanova, Elena Apostolova, Tonka Vasileva, Mariana Nikolova, Ivanka Dimitrova-Dyulgerova, Tzenka Radoukova

**Affiliations:** Faculty of Biology, Plovdiv University, 24 Tsar Asen Street, 4000 Plovdiv, Bulgaria; mollov_i@uni-plovdiv.bg (I.M.); gyahubyan@uni-plovdiv.bg (G.Y.); mariank@uni-plovdiv.bg (M.G.); eapostolova@uni-plovdiv.bg (E.A.); vasileva@uni-plovdiv.bg (T.V.); mariana.nikolova@uni-plovdiv.bg (M.N.); ivadim@uni-plovdiv.bg (I.D.-D.); radoukova@uni-plovdiv.bg (T.R.)

**Keywords:** *Fontinalis antipyretica*, rbcL, PCR, TPC, leaf micromorphological characteristics

## Abstract

**Simple Summary:**

Pollution of the aquatic environment is a well-known problem with a long history. Monitoring water quality relies on biota in order to provide adequative assessment and management of the water bodies. Among the different biological indicators applied, aquatic macrophytes, and particularly mosses, are in direct relationship with the environment and their use as biomonitors is well documented. In the current study, we made an attempt to apply new fast, reliable and comprehensible methods for water pollution control. Three reservoirs were selected for the following reasons: (i) they were polluted with hazardous substances (heavy metals and organic material) and (ii) they are used for fish farming and irrigation and their water quality directly affects human health. Moss-bags with the selected biomonitor *Fontinalis antipyretica* were exposed in the reservoirs for a period of 30 days and molecular, chemical and micromorphological markers were studied. All biomarkers tested appeared to be sensitive to the pollution. This research provided a basis for further studies on selected biomarkers towards standardization.

**Abstract:**

Moss-bags were applied to study the effect of contamination in three standing water bodies in Bulgaria (Kardzhali, Studen Kladenets and Zhrebchevo Reservoirs), the first two with old industrial contamination and the last polluted with short-chain chlorinated paraffins (SCCPs). *Fontinalis antipyretica* Hedw. collected from background (unpolluted) site was placed in cages for a period of 30 days. The present study examined whether inorganic and organic pollution detected with moss-bags resulted in corresponding differences in molecular, chemical and micromorphological markers. Suppressed large subunit of ribulose-1,5-bisphosphate carboxylase (rbcL) expression was assessed in moss-bags from two of the reservoirs, contaminated with heavy metals. There was a decrease of the total phenolic content (TPC) in the moss-bags, which provides a basis for further studies of the chemical content of aquatic mosses. *Fontinalis antipyretica* also showed a response through leaf micromorphological characteristics. In the all three reservoirs, an increase of the twig leaf cell number was recorded (*p* ≤ 0.01 for Kardzhali and *p* ≤ 0.001 for Studen Kladenets and Zhrebchevo reservoirs), as well as of the stem leaf cell number in Zhrebchevo Reservoir (*p* ≤ 0.001). On the contrary, the width of the cells decreased in the studied anthropogenically impacted reservoirs. All three studied groups of biomarkers (molecular, chemical and micromorphological) appeared to be sensitive to freshwater pollution. The results achieved indicated that rbcL gene expression, TPC, cell number and size are promising biomonitoring tools.

## 1. Introduction

Biomarkers represents a quantitative measure of cellular and subcellular exposure response, as well as response of processes, structures and functions to stress [1]. Such a response can be established via chemical and enzyme activities alteration, damage to proline content and leaf micromorphological characteristics as a result of heavy metal and other pollutants [2].

Gene expression appeared to be a promising biomarker in terms of economy in application and significance of results [1]. Moreover, the response at the molecular level is within minutes to an hour [3]. Plant ribulose-1,5-bisphosphate carboxylase/oxygenase, known as RuBisCO, is the major enzyme involved in the atmospheric fixation of carbon dioxide in the organic carbon required for plant growth and development [4,5]. However, RuBisCo can accept oxygen as a substrate and catalyze an oxygenation reaction that initiates the photorespiratory pathway in which plants lose energy [6]. RuBisCo is represented by eight large and eight small subunits, and its activity depends on RuBisCo activase [7]. The evaluation of changes in the catalytic efficiency of RuBisCo in cases of heavy metal contamination is important due to its ability to affect the net rate of photosynthesis [8]. Decreased RuBisCo activity was reported in several studies: in *Phaseolus vulgaris* under Zn^2+^ stress [9], in *Erythrina variegata* under stress from Cd [10], in *Chenopodium rubrum* [11] and in *Helianthus annuus* due to Cu^2+^ exposure [12]. To cope with Cd stress, *A. thaliana* increases the abundance of RuBisCO-binding subunit proteins (LSUs) [13]. According to Pankovic et al. [14], the inhibitory effect of Cd on sunflower photosynthesis can be alleviated by providing an optimal level of N nutrition, which significantly improves the performance of RuBisCo. The amount and activity of RuBisCo/RuBisCo activase in tobacco plants stressed with Cd/Cu were improved by treatment with sodium nitroprusside (SNP), which suggests a protective role of SNP against toxic damage caused by heavy metals [15].

Antioxidant compounds in plants are used to protect cells from oxidative damage [16]. These compounds are mainly phenolic and include carotenoids, tocopherols, phenolic acids (cinnamon acids and benzoic acid derivatives), dipropenes and flavonoids [17].

Plant morphological and anatomical characteristics also respond to environmental variations. Epidermal features like number of leaf epidermis basic cells [18,19] and stomata [19,20,21] in terrestrial vascular plant species demonstrated sensitivity to environmental stress.

According to the latest review on biomarkers in aquatic plants, researches in this area are limited [22]. The most commonly used moss in water quality monitoring, *Fontinalis antipyretica* is widespread throughout the northern hemisphere and is the most easily recognized [23]. The selected species can be successfully employed as a biomonitor in water bodies where it does not naturally present [24]. An exposition to Cd, Cu, Pb and Zn led to increased levels of superoxide dismutase (SOD), catalase (CAT), glutathione reductase (GRD), ascorbate (APX) and guaiacol peroxidases (GPX) in *Fontinalis antipyretica* [25], as well as a correlation between polycyclic aromatic hydrocarbons (PAHs) and antioxidant enzymes was reported [26].

In the frame of the present study, alive specimens of *Fontinalis antipyretica* from background site were exposed at three reservoirs in Bulgaria, among them two with industrial contamination and one affected by untreated wastes. The current study aimed to analyze the effect of pollutants on molecular markers, antioxidant potential and amount of total phenolic content, as well as on leaf micromorphological characteristics as novel stress biomarkers. As far as the accomplished literature survey could ascertain, the above biomarkers have not previously been studied simultaneously. Hence, we hypothesized that they are influenced by environmental factors, specifically by contaminants and could be employed to monitor aquatic environment pollution.

## 2. Materials and Methods

### 2.1. Field Collection and Moss-Bags

Moss-bags with *Fontinalis antipyretica* collected from a small unpolluted stream in Bulgaria (42.119625° N, 24.556055° E WGS 84), were placed at three reservoirs for 30 days during the summer of 2019. Each moss-bag contained plant material of about 100 g wet weight in a flat bag (30 × 20 cm) from plastic mesh (aperture 1 mm). At the end of the position period, moss was rinsed on-site with the reservoir water and after removal of material in poor condition and foreign objects was transported in coolers (2–4 °C) to the laboratories.

The selected reservoirs are affected by a different degree of pollution [24]. Kardzhali (41.638475° N, 25.304432° E WGS 84) and Studen Kladenets (41.622244° N, 25.441933° E WGS 84) reservoirs suffer from old industrial contamination and have sediments’ loads of priority substances (e.g., Cd and Pb). Zhrebchevo Reservoir (42.585571° N, 25.885592° E WGS 84) is located in a region with light industries but is under the influence of illegal dumping sites (solid waste) and untreated wastewaters. Among the studied reservoirs, As, Cd, Co, Cu, Mn, Ni, P, Pb, Zn had maximum levels in the water from Studen Kladenets, Al and Cr in Kardzhali Reservoir, as well as both reservoirs had maximum of polybrominated diphenyl ethers (PBDE) congeners [24]. Only short-chained chlorinated paraffins (SCCPs) had maximum in water samples from Zhrebchevo Reservoir, above the environmental quality standard (EQS).

### 2.2. Nucleic Acid Extraction

Moss samples were dried immediately on filter paper at room temperature, immersed in liquid nitrogen, and stored in a freezer at −80 °C until the moment of DNA extraction. The expression analysis was performed in three technical and three biological replicates. Total DNA was extracted using DNeasy Plant Mini Kit (Qiagen) and total RNA was extracted using RNeasy Plant Mini Kit (Qiagen) according to the manufacturer’s instructions. The quality and quantity of extracted nucleic acids were assessed on the Epoch microplate spectrophotometer and checked on 1% agarose gel.

### 2.3. PCR and RT-qPCR

PCR reactions were performed in a reaction volume of 20 µL using 1µL of DNA (50 ng/µL); 1 µL deoxynucleotide triphosphates—dNTPs (5 mM), 2.5 µL Buffer (10×), 1.5 µL MgCl2 (25 mM), 0.5 µL Fw/Rev Primer (10 µM), 0.13 µL Taq polymerase (5 U/µL) and Nuclease-free H20 to final volume. The amplification was performed using a Biosystems 2720 Thermal Cycler. The amplification program was started at an initial stage of 4 min at a temperature of 94 °C, followed by 30 cycles: denaturation at 94 °C for 30 s, addition of primers at a temperature of 56 °C for 30 s, and elongation at 72 °C for 60 s. A final step of 72 °C for 7 min was added to the PCR program, necessary for the final elongation of the amplification products. The resulting products were divided into 1% agarose gel.

To check the relative expression of rbcL gene, 1 µg RNA was reverse transcribed with Script cDNA polymerase (Jena Bioscience, Germany) following the manufacturer’s instructions. EF1a gene was used as an endogenous control to normalize the expression level of the rbcL gene.

RT-qPCR reactions were performed in three technical replicates on an Applied Biosystems 7500 Real-time PCR System using the Green Master Mix Kit (Genaxxon Bioscience). Finally, a volume of 25 μL was used to perform the PCR amplification reaction. Copy the DNA template was diluted to the appropriate concentration. Relative quantitation of gene expression (RQ) was determined by the 2^−ΔΔCT^ method [27].

### 2.4. Sequencing

PCR products were sequenced by Eurofins Genomics GmbH, Ebersberg, Germany.

### 2.5. Non-Enzymatic Antioxidant Activities: Preparation of Aquatic Moss Extracts

The sample (10 g frozen aquatic moss, two replicates) was mixed with a composite of organic solvent (acetone):water:acid (70/30/1 *v*/*v*/*v*). The selected acid was hydrochloric acid. The conditions for the extraction were as follows: solid/liquid ratio 1:8, on a magnetic stirrer at temperature of 25 °C for 60 min and filtered through nylon cloth (double extraction). The collected organic extracts were concentrated to dry substance using a rotary evaporator at a temperature not exceeding 50 °C.

### 2.6. Total Phenolic Content (TPC)

The Folin–Ciocalteu method was used to determine TPC as described by Singleton and Rossi [28]. The absorbance readings were taken at 760 nm using UV–VIS spectrophotometer DU 800 (Beckman Coulter ^®^, Brea, CA, USA) after incubation for 5 min at 50 °C. Gallic acid was used as reference standard. The results were expressed as milligram gallic acid equivalent per 100 g sample (mg GAE/100 g sample).

### 2.7. Antioxidant Activity Assay

The antioxidant activities of the samples were determined using DPPH and CUPRAC methods. Radical scavenging activity (RSA) of extracts against 2,2-diphenyl-1-picrylhydrazyl (DPPH) radical was assessed spectrophotometrically at 517 nm. The assay was done according to a method reported by Brand-Wiliams et al. [29]. A DPPH solution (80 μM) was freshly prepared in methanol. A volume of 2 mL of this solution was allowed to react with 150 μL of sample extracts and the absorbance was measured after 30 min to 180 min in dark. The antioxidant activity was calculated as follows:(1)$$RSA\;\left(\%\right)=\frac{\left(A_{\mathrm{control}}-A_{\mathrm{sample}}\right)}{A_{\mathrm{control}}}\times \;100$$
where A_control_ is the absorbance of the DPPH radical in methanol; A_sample_—the absorbance of the DPPH radical solution mixed with a sample of aquatic moss *Fontinalis antipyretica*.

The CUPRAC assay was performed as described by Apak et al. [30]. The method involves mixing the solutions of 10 mM CuCl2, 7.5 mM neocuproine, 1 M ammonium acetate at pH 7, and 50 μL of different extracts and measuring the absorbance at 450 nm after 60 minutes.

The final absorbance was compared with the standard curve in the range of 0 mM to 0.5 mM Trolox, dissolved in methanol. The data were expressed as mM Trolox equivalent/g sample (mM TE/g sample).

### 2.8. Leaf Micromorphological Characteristics

Samples of *Fontinalis antipyretica* were fixed in 70% ethanol. Mature leaves from the stem (stem leaves) and from lateral twigs (twig leaves) were observed under light Magnum T Trinocular microscope CETI (Medline Scientific, Chalgrove, UK). The number, length and width of the cells were examined under a magnification of 400 times. Fifty measurements were made for each characteristic per station. Photomicrographs were taken using digital camera Si 5000 5 Mpx (Medline Scientific, Chalgrove, UK).

### 2.9. Statistical Analysis

Descriptive statistics was applied to process the results obtained for leaf micromorphological characteristics. Student’s *t*-test for independent samples was used to determine the differences between the experimental and control data, which were normally distributed. The chosen levels of significance (*p*) were *p* < 0.05, *p* < 0.01 and *p* < 0.001. Statistical processing of the data was done using the computer software PAST [31].

## 3. Results

### 3.1. Expression Analysis of the Chloroplast rbcL Gene (Ribulose 1,5-Bisphosphate Carboxylase/Oxygenase Large Subunit)

In the rbcL expression assay, elongation factor 1-α (EF1-α) was used as an endogenous control in RT-qPCR. Since the gene sequence of *Fontinalis antipyretica* is not available in databases, we carried out alignment of plant EF1-α to find a region conserved at both the DNA level and the amino acid sequence level. The amplification of *Fontinalis antipyretica* genomic DNA with degenerate primers flanking the gene intron yielded two bands that were cloned and sequenced. The deduced amino acid sequence revealed high similarity to the plant EF1-α for one of the amplicons. The most significant alignment was identified with EF1-α of the moss *Physcomitrium patens*, which is a model organism for non-seed plants (Figure 1).

The highest relative abundance of rbcL was observed in the moss from Zhrebchevo Reservoir (Figure 2). Compared to it, the gene expression was much lower in the samples collected from Kardzhali and Studen Kladenets reservoirs.

### 3.2. Non-Enzymatic Antioxidant Activities

As secondary metabolites polyphenols had lower values in the transplants than in the background moss (Table 1). The lowest antioxidant activity represented as RSA (54.9%) was observed in the moss-bags from Kardzhali Reservoir, while following the CUPRAC method, the lowest antioxidant activity was detected in moss from Zhrebchevo Reservoir. The applied two methods are based on different mechanisms, including hydrogen atom transfer (DPPH) and single electron transfer (CUPRAC). Moreover, antioxidant activities largely depend on the content of polyphenolic substances, both quantitatively and qualitatively (type of polyphenolic compounds). It could be suggested that low values of non-enzymatic antioxidant activity illustrated that moss cannot compensate the pollution effect in a non-enzymatic way.

Both antioxidant methods (DPPH, CUPRAC) did not show correlation with TPC.

### 3.3. Leaf Micromorphological Characteristics

The leaf lamina of the *Fontinalis antipyretica* gametophyte consists of one-layer elongated photosynthesizing cells with straight anticlinal cell walls (Figure 3).

#### 3.3.1. Number of Cells/mm^2^

Data from microscopic analysis showed a higher number of cells per 1 mm^2^ in the twig leaves than in the stem leaves (Table 2, Figure 3). Moss from the background site had the lowest mean values for this biomarker, both for the stem (918.5/mm^2^) and the twig leaves (1153.1/mm^2^). In the reservoirs, under anthropogenic impact, an increase in the number of leaf cells was recorded (between 935.9/mm^2^ and 952.2/mm^2^ for the stem leaves and 1195.3/mm^2^ and 1227.4/mm^2^ for the twig leaves). However, the difference from the background was statistically significant for the twig leaves (*t*-test, *p* < 0.01 for Kardzhali and *p* < 0.001 for Studen Kladenets and Zhrebchevo Reservoir), while for the stem leaves—only in Zhrebchevo Reservoir (*t*-test, *p* < 0.001; Table 2). The calculated coefficient of variation for the number of cells was in the range of 2.6–8.6% (Table 2).

#### 3.3.2. Cell Dimensions

Cell length in stem leaves was greatest in the background samples (95.5 µm mean; 75 µm min; 135 µm max), followed by Kardzhali and Studen Kladenets reservoirs, where the differences were not statistically significant and only in Zhrebchevo Reservoir did the cells differ of shorter length (85.3 µm mean; 56 µm min; 119 µm max; *t*-test, *p* ≤ 0.01). In twig leaves, the cells were shorter in length than in stem leaves, an exception was the moss from Zhrebchevo Reservoir, where the length of the stem and twig leaves was almost similar (85.3 µm and 80.1 µm, respectively). Cell length in twig leaves showed the same tendency to decrease in the samples from Zhrebchevo Reservoir compared to the background (Table 2, Figure 3). Cell width both in the background stem and twig leaves was 11.8 µm and 8.1 µm, respectively. In the anthropogenically affected reservoirs, the width of the cells decreased, except for the stem leaves from the Kardzhali Reservoir (Table 2). The coefficient of variation had higher values for cell sizes (9.1–22.1%) compared to their number (Table 2).

## 4. Discussion

### 4.1. Molecular Biomarker

RbcL is one of the two subunits of RuBisCO, which is encoded by chloroplast DNA in plants. Recently, it was shown that the expression of rbcL gene can be used as a biomarker to monitor plant stress response to complex heavy metal contaminations [32]. Heavy metal pollution has a detrimental effect on photosynthetic metabolism including restrained carboxylation efficiency of RuBisCo [33]. Previous research on water quality revealed severe heavy metal contamination in Kardzhali and Studen Kladenets reservoirs [23]. In our study, the suppressed rbcL expression in the moss-bags from Kardzhali and Studen Kladenets reservoirs was most likely a consequence of the increased content of heavy metals and can affect negatively the photosystem activity.

### 4.2. Chemical Biomarkers

The available literature on the chemical content of *Fontinalis antipyretica* is scarce. Comparison with TPC (6.25–18.21 mg/g) of the terrestrial moss *Hypnum cupressiforme* from low mountain area in Serbia [34] showed higher TPC in the aquatic moss from the background station. The TPC decreased in the moss over exposure period in all selected reservoirs, while antioxidant activity had no clear trend of gradual decrease in polluted aquatic environment. Further studies on phenolic content and possible impact on it of water pollution could be recommended.

### 4.3. Micromorphological Biomarkers

Increased number of cells and decreased cell size, mainly in width, was registered in *Fontinalis antipyretica* exposed at all three reservoirs. The tested micromorphological biomarkers showed response in conditions of heavy contamination with Al, As, Cd, Cr, Ni and Pb (Kardzhali Reservoir), with Al, Cd, Cr, Cu and Pb (Studen Kladenets Reservoir), with SCCPs (Zhrebchevo Reservoir) and PBDEs in all three reservoirs [24]. Nevertheless, the most pronounced was the alteration in the above biomarkers in the moss samples from Zhrebchevo Reservoir, which could be linked with the assessed short-chain chlorinated paraffins (SCCPs) in water above the EQS and a maximum of 9.2 µg kg^−1^ accumulated in the moss-bags. The leaves from the twigs appeared to be more sensitive than the stem leaves. Based on the results achieved, the number and size of cells in the twig leaves could be suggested as reliable biomarkers to enable early recognition of pollution stress, especially in response to organic priority substances.

Our results also confirmed that the increasing number of cells, especially epidermal, associated with a decrease in their size are among the most indicative micromorphological leaf features for environmental pollution [18,19].

## 5. Conclusions

The applied moss-bags and biomarkers studied have proven to be sensitive in terms of polluted aquatic environment. The research provided an insight into chloroplast rbcL gene expression of the aquatic biomonitor *Fontinalis antipyretica* transplanted in Bulgarian reservoirs. The suppressed rbcL expression in the moss-bags from two reservoirs under former industrial impact was most likely a consequence of the severe contamination with heavy metals and can affect negatively the photosystem activity. The TPC in moss-bags diminished in the environmental conditions of the three reservoirs but the results achieved need further confirmation and provide a basis for it. The leaf micromorphological characteristics in *Fontinalis antipyretica* also reflected the deteriorated water quality: cell number increased, while cell size decreased. A solution within the frame of freshwater pollution cannot be offered by a biomarker itself, but a multi-parametric approach can contribute to the assessment and management of the water bodies.

## Figures and Tables

**Figure 1 biology-10-00003-f001:**
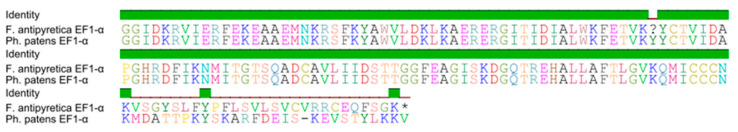
Deduced amino acid sequence of *Fontinalis antipyretica* elongation factor 1-α (EF1-α) compared to *P. patens* EF1-α.

**Figure 2 biology-10-00003-f002:**
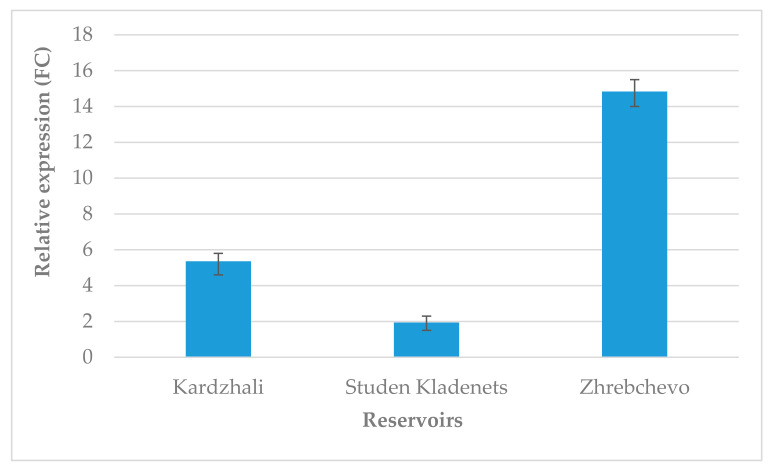
Response of the chloroplast rbcL gene (ribulose 1,5-bisphosphate carboxylase/oxygenase large subunit) in *Fontinalis antipyretica*. Expression was evaluated by reverse transcriptase polymerase chain reaction (RT-qPCR), normalized to EF1-α and presented as fold change (FC). Representative experiment from three independent replicates is shown. Error bars indicate SD of replicates.

**Figure 3 biology-10-00003-f003:**
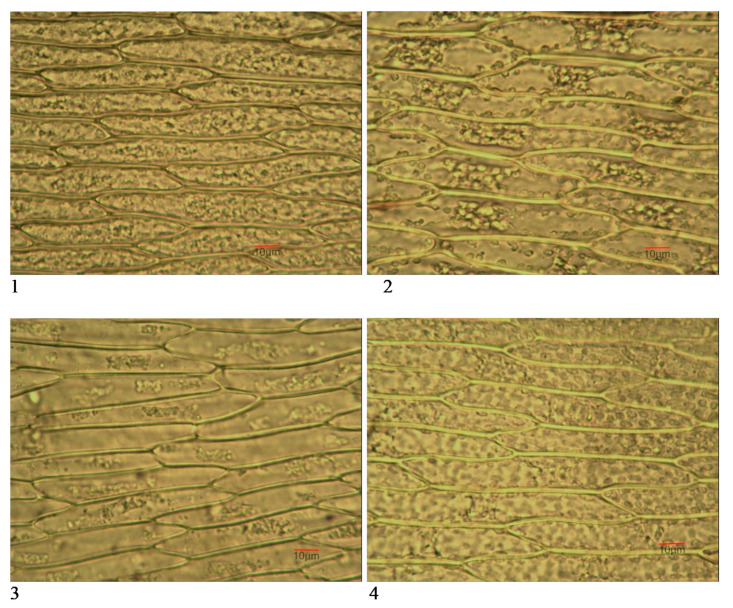
Photomicrographs of the leaf cells from *Fontinalis antipyretica*: **1**,**2**—Background (Control); **3**,**4**—Zhrebchevo Reservoir; **1**,**3**—twig leaves; **2**,**4**—stem leaves.

**Table 1 biology-10-00003-t001:** Antioxidant activity and polyphenolic compounds in *Fontinalis antipyretica*. Legend: TPC—total phenolic content, RSA—radical scavenging activity, CUPRAC—CUPric Reducing Antioxidant Capacity.

Sample	TPC mg GAE/100 g	RSA %	CUPRAC Assay mM TE/g
Background	49.9 ± 1.6	77.9	1.45 ± 0.1
Kardzhali	29.3 ± 1.8	54.9	2.44 ± 0.14
Studen Kladenets	27.3 ± 0.4	80.2	1.34 ± 0.08
Zhrebchevo	29.0 ± 0.4	74.2	0.85 ± 0.05

**Table 2 biology-10-00003-t002:** Micromorphological cell characteristics of *Fontinalis antipyretica*. Legend: X—arithmetic mean; min—minimum; max—maximum value; S_x—_arithmetic mean error; CV—coefficient of variation; cell length and width are given in µm; level of significance: * *p* ≤ 0.05; ** *p* ≤ 0.01; *** *p* ≤ 0.001.

Leaf Samples		Stem Leaves			Twig Leaves		
		min (X ± S_x_) max	S_x _%	CV %	min (X ± S_x_ ) max	S_x_ %	CV %
Background	Number of cells/mm^2^	851 (918.5 ± 7.6) 989	0.8	4.5	989 (1153.1 ± 12.5) 1265	1.1	5.9
	Cell length	75 (95.5 ± 2.5) 132	2.6	14.6	66 (87.2 ± 2.6) 114	3	16.7
	Cell width	9 (11.8 ± 0.2) 14	1.8	10.1	5 (8.1 ± 0.2) 11	2.4	13.9
Kardzhali	Number of cells/mm^2^	849 (935.9 ± 15.8) 1173	1.6	8.6	1111 (1195.3 ** ± 11.6) 1295	0.8	4.5
	Cell length	61 (92.5 ± 3.4) 120	3.6	20	62 (86.3 ± 2.3) 111	2.6	14.4
	Cell width	9 (11.4 ± 0.2) 14	1.5	9.1	5 (7.5 * ± 0.2) 10	2.1	12.1
Studen Kladenets	Number of cells/mm^2^	874 (946.1 ± 14.1) 1127	1.4	8.1	1142 (1204 *** ± 6.2) 1257	0.4	2.6
	Cell length	70 (94.2 ± 2.9) 127	3	16.7	69 (84.9 ± 2.9) 122	3.4	18.8
	Cell width	8 (10.8 ** ± 0.26) 14	2.4	13.2	5 (7.3 ** ± 0.2) 11	2.3	12.5
Zhrebchevo	Number of cells/mm^2^	851 (952.2 *** ± 6.8) 1012	0.7	3.9	1127 (1227.4 *** ± 12.1) 1357	0.9	5.4
	Cell length	56 (85.3 ** ± 3.4) 119	4	22.1	61 (80.1 * ± 2.2) 110	2.7	15
	Cell width	7 (10.7 ** ± 0.3) 15	2.9	16.4	4 (7.4 ** ± 0.2) 10	2.9	16.4

## Data Availability

Not applicable.

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
