# Peer review of "Can Biomarkers Respond Upon Freshwater Pollution?—A Moss-Bag Approach"

_biology, 2020, doi:10.3390/biology10010003_

Round 1

Reviewer 1 Report

Comments on manuscript: Can biomarkers reflect the freshwater pollution? (biology-1024430)

The manuscript proposes some potential biomarkers of freshwater pollution and helps providing additive assessment and management of the water bodies. The manuscript would be of some interest for the audience of Biology. However, there are general problems in the manuscript that have to be considered. Detailed comments are as follows.

Title: I don’t think the title is fully reflect the study, especially the use of moss bags

Abstract

No conclusion mentioned in the abstract, only methods and results seem not so adequate to demonstrate the study.

Line 41: Why “Transplanted” here, abovementioned were moss-bags. For the moss-monitoring studies, both transplanted and moss-bag have been applied, they are different approaches.

Introduction:

The background of the study was quite unclear, as well as the logic, not just list several of the pollutant or physiological indices. What are the biomarkers? Are there any limitations in studies of the freshwater pollution? What are the relationships between biomarkers and aquatic pollution? The moss species Fontinalis antipyretica is widespread, then why you used moss-bags, but not natural growing mosses?

Materials and Methods

Materials and methods are not very clear. What’s the size of the moss bags and how about the biomass of the moss in a bag? A detailed description should be present about the moss-bags. And whether the mosses would be saturated with pollutants. How did you ascertain this unpolluted stream? Did you measure the pollutants in the stream?

Why did you use both DPPH and CUPRAC methods to determine antioxidant activities, are there differences or compensation of these two methods, and how to evaluate the results of the two methods? These should be clarified.

About the statistical analysis, since the author analyzed three reservoirs, the differences among them should be quantified, as well as related results showing.

Line 89-91: the unit of ° were missed in the longitudes and latitudes. And What are the priority substances?

Line 140-141: the meaning of A_control, A_sample, and CUPRAC should be given.

Results

Full names of abbreviations in the tables and figure should be given for the first time, since they should be readable beyond of the main text.

No statistical analysis conducted on the non-enzymatic antioxidant activities. Also, RSA and CUPRAC assay showed different trends from background to Zhrebchevo, are there any relations of these two methods, and how to quantify the pollution according to the results of antioxidant activity?

Line 177 and Figure 2: Error bars weren’t showed in Figure 2, please add.

In the caption of Table 2, the meaning of * should be given more specifically, and % should be removed?

Discussion and Conclusions:

Could you give us some explanations on the differences of chemical and micromorphological biomarkers in three reservoirs, the probabilities of these indices used as biomarkers of water pollution, and how to assess the water quality by using these biomarkers?

Author Response

Dear Reviewer 1,

Thank you very much for the constructive review and for the opinion that the study proposes some potential biomarkers of freshwater pollution and helps providing additive assessment and management of the water bodies. Your concerns were addressed and below you can find our answers to your comments. (Reviewer’s comments are shown in Italic).

Title: I don’t think the title is fully reflect the study, especially the use of moss bags

Thanks to your suggestion, the title was revised: “Can biomarkers in moss-bags respond upon freshwater pollution?”.

Abstract

No conclusion mentioned in the abstract, only methods and results seem not so adequate to demonstrate the study.

Thanks to your suggestion, conclusion sentences were added (Line 48-50).

Line 41: Why “Transplanted” here, abovementioned were moss-bags. For the moss-monitoring studies, both transplanted and moss-bag have been applied, they are different approaches.

We agree with the Reviewer, “transplanted” was removed (Line 43).

Introduction:

The background of the study was quite unclear, as well as the logic, not just list several of the pollutant or physiological indices. What are the biomarkers? Are there any limitations in studies of the freshwater pollution? What are the relationships between biomarkers and aquatic pollution? The moss species Fontinalis antipyretica is widespread, then why you used moss-bags, but not natural growing mosses?

We thank the Reviewer 1 for this comment. Following Your and Reviewer 2 recommendations, the text in the Introduction was reorganized and connected in a logical order. Important literature was included (e.g. Bartell 2006, Akhtar et al. 2005, etc.) in order to reflect the above questions (Line 55-60, Line 85-86).

Materials and Methods

Materials and methods are not very clear. What’s the size of the moss bags and how about the biomass of the moss in a bag? A detailed description should be present about the moss-bags. And whether the mosses would be saturated with pollutants. How did you ascertain this unpolluted stream? Did you measure the pollutants in the stream?

Thanks to your suggestion and those of Reviewer 2, more detailed information about the study design was provided (Please see Materials and Methods, Line 106-110). Each moss-bag contained about 100 g wet weight of the moss.

The selection of the background station was made based on monitoring data of the East Aegean Water Basin Directorate, Ministry of Environment and Water, Bulgaria and was ascertained by the results received from inorganic and organic analyses (twenty-four compounds). The concentrations in water samples from the background station were below the LOD for Hg (<0.05 µg L-1) and Fe (<0.1 mg L-1), all of the rest 14 analyzed elements had minimum levels at the background station. All studied PBDE congeners and SCCPs were below the LOD in water samples from the background station, except for BDE-99 and 100 (as stated in Gecheva et al., 2020).

Why did you use both DPPH and CUPRAC methods to determine antioxidant activities, are there differences or compensation of these two methods, and how to evaluate the results of the two methods? These should be clarified.

Two methods (DPPH and CUPRAC) were used to determine the non-enzymatic antioxidant activity of aquatic mosses extracts in order to receive a deeper characterization of the biomarker. They both are quick and easy to be implemented, but employ different mechanisms for evaluation. Unfortunately, the lack of similar studies evaluating the antioxidant properties of moss extracts, impede the direct comparisons at that stage, but we hope that the results reported will create a beginning of a database. An attempt to shed light on different trends, relation between the methods and possible interpretation of the data was made (Please see Line 218-223).

About the statistical analysis, since the author analyzed three reservoirs, the differences among them should be quantified, as well as related results showing.

Thanks to your recommendation, the subsection 2.9. Statistical analysis was clarified (Line 185-188), as well as the method applied for the evaluation of the differences was added in subsections 3.3.1 and 3.1.2.

Line 89-91: the unit of ° were missed in the longitudes and latitudes. And What are the priority substances?

Thank you – the unit of ° was added. The term priority substances was used in the context of Directive 2013/39/EU of the European Parliament and of the Council. The substances referred in the sentence pointed by you, are in general Pb and Cd. (Line 114)

Line 140-141: the meaning of A_control, A_sample, and CUPRAC should be given.

The meaning of the compounds in the equation was given (Line 168-170).

Results

Full names of abbreviations in the tables and figure should be given for the first time, since they should be readable beyond of the main text.

Full names of abbreviations in the tables and figures were given: in the captions of Figure 1, Figure 2, Table 1.

No statistical analysis conducted on the non-enzymatic antioxidant activities. Also, RSA and CUPRAC assay showed different trends from background to Zhrebchevo, are there any relations of these two methods, and how to quantify the pollution according to the results of antioxidant activity?

An attempt to shed light on different trends, relation between the methods and possible interpretation of the data was made (Please see Line 218-223).

Line 177 and Figure 2: Error bars weren’t showed in Figure 2, please add.

Thank you – the error bars were added.

In the caption of Table 2, the meaning of * should be given more specifically, and % should be removed?

Thank you - The meaning of * was given and % was removed.

Discussion and Conclusions:

Could you give us some explanations on the differences of chemical and micromorphological biomarkers in three reservoirs, the probabilities of these indices used as biomarkers of water pollution, and how to assess the water quality by using these biomarkers?

Thanks to your comments, a more detailed information about the response of micromorphological biomarkers applied was added (Please see Line 279-288).

Yours sincerely,

Gana Gecheva and the co-authors

Reviewer 2 Report

This work is interesting as it evaluates the application of new techniques to pollution studies using aquatic moss as a biomonitor. But it is inconsistent, given that basic information is missing. This work is linked to another work published in the journal Water, in which much of the information missing in this article is given. The inclusion of that information, and the corresponding analysis of it, would give another consistency to this article.

In addition, the article needs to be improved in the following aspects:

  • Authors must be careful with the use of acronyms, the full name must always be indicated the first time they are cited in the text. Another option is to include a specific section of Abbreviations.
  • The introduction should be rewritten, relating the information given in the different paragraphs to each other and to the objectives of the article, and providing more information on the state of the art about the markers analyzed in this work.
  • The materials and methods section must be rewritten. Important methodological information is missing, as well as more references about the methods used.
  • The discussion and conclusions are quite speculative. The results provided are not consistent enough to answer the question that gives the title to the article, nor the objectives set out in the summary and introduction. It cannot be concluded that the physiological and morphological parameters measured in the moss are indicators of pollution, when the data of contaminants are not provided. It would be necessary to perform statistical analyzes to determine the relationship between pollutants and biomarkers.

Other more specific comments are included in the article pdf.

Author Response

Dear Reviewer 2,

We sincerely thank the Reviewer 2 for the evaluation of the manuscript and valuable comments. We appreciate the inputs given, which definitely helped to improve our manuscript. Below you can find our answers to your specific comments. (Reviewer’s comments are shown in Italic).

This work is interesting as it evaluates the application of new techniques to pollution studies using aquatic moss as a biomonitor. But it is inconsistent, given that basic information is missing. This work is linked to another work published in the journal Water, in which much of the information missing in this article is given. The inclusion of that information, and the corresponding analysis of it, would give another consistency to this article.

The Reviewer brings up an important point. As the Reviewer 2 mentioned above, the results achieved are received within the frame of a large-scale study. The published and cited material in the Water Journal (Gecheva et al., 2020) was focused on the applicability of moss-bags and mussel transplants as bioaccumulative monitors. The current manuscript was an attempt to apply a multi-parametric approach based on moss biomarkers’ response in confirmed pollution. First of all, we decided to focus the research on discovering novel and reliable biomarkers, secondly, we have no right to include data that have already been published, thirdly in our opinion the focus here is on the response of biomarkers, not on the pollutant levels accumulated (as in Gecheva et al., 2020).

In general, we followed the scheme of previous extended researches as those of Say and Whitton, 1983; Wehr and Whitton, 1983a, 1983b, published in Hydrobiologia Journal under the project “Using aquatic plants to monitor water quality”.

Following the Reviewers 2 concern, data for analyzed pollutants were provided (Please see Line 116-119). In addition, a more detailed information was provided the manuscript, e.g. about the response of micromorphological biomarkers (Please see Line 279-288).

Authors must be careful with the use of acronyms, the full name must always be indicated the first time they are cited in the text. Another option is to include a specific section of Abbreviations.

We agree with Reviewer 2. Following both Reviewers’ recommendations, the full names of acronyms were given the first time they were cited in the text (in the captions of Figure 1, Figure 2, Table 1, Abstract – rbcL, Introduction - SOD, CAT, GRD, APX, GPX, PAHs, 2.3. – dNTPs).

The introduction should be rewritten, relating the information given in the different paragraphs to each other and to the objectives of the article, and providing more information on the state of the art about the markers analyzed in this work.

We thank the Reviewer for this comment. Following this recommendation, as well as those of Reviewer 1, the text in the Introduction was reorganized and connected in a logical order. Important literature was included (e.g. Bartell 2006, Akhtar et al. 2005, etc.) in order to reflect the above questions (Line 55-60, Line 85-86).

The materials and methods section must be rewritten. Important methodological information is missing, as well as more references about the methods used.

Thanks to your suggestion and those of Reviewer 2, more detailed information about the study design was provided (Please see Materials and Methods, Line 106-110).

The discussion and conclusions are quite speculative. The results provided are not consistent enough to answer the question that gives the title to the article, nor the objectives set out in the summary and introduction. It cannot be concluded that the physiological and morphological parameters measured in the moss are indicators of pollution, when the data of contaminants are not provided. It would be necessary to perform statistical analyzes to determine the relationship between pollutants and biomarkers.

Thank you for this valuable comment. The title of the article was revised. As stated in the manuscript the current study aimed to analyze the effect of pollutants on molecular markers, antioxidant potential and amount of total phenolic content, as well as on leaf micromorphological characteristics (Line 95-97). As stated above on a previous answer to Your comment (i) we focus the research on discovering novel and reliable biomarkers; (ii) we have no right to include data that have already been published; (iii) the focus here is on the response of biomarkers, not on the pollutant levels accumulated (as in Gecheva et al., 2020) and (iv) we followed the scheme of previous extended researches as those of Say and Whitton, 1983; Wehr and Whitton, 1983a, 1983b, published in Hydrobiologia Journal under the project “Using aquatic plants to monitor water quality”.

Following the Reviewers 2 concern, data for analyzed pollutants were provided (Please see Line 116-119).

Specific comments are included in the article pdf.

Your comments in the pdf were addressed and below you can find our answers to specific comments. PCR was added as a keyword.

In this study it does not analyze pollutants, or indicate data on pollutant type, concentrations, etc. Therefore, it is difficult to establish such a relationship.

Due to the importance of this issue, as was written above, data for analyzed pollutants were provided (Please see Line 116-119) in order to establish relationship.

The explanation about this biomarker is very detailed, and as it is presented it is cumbersome. It would be clearer if the results of the different studies cited were summarized. It also contrasts with the brevity with which the other biomarkers are explained. Also, part of this results can be used in the discussion

Following Your suggestion, a paragraph in the Introduction section was summarized.

More explanation about the procedure used is needed: preparation of the moss, quantity and part of the moss used, characteristics of the bags .....

Thanks to Your and Reviewer 1 suggestion, more detailed information about the study design was provided (Please see Materials and Methods, Line 107-110). Each moss-bag contained about 100 g wet weight of the moss.

Which one? You can indicate? And their environmental characteristics?

Thank you – coordinates were added, please see Line 106.

Specify more. What substances?

The term priority substances was used in the context of Directive 2013/39/EU of the European Parliament and of the Council. The substances referred in the sentence pointed by you, are in general Pb and Cd. (Line 114). Data for analyzed pollutants were provided Line 116-119.

Relevant methodological information is missing. For example regarding: transport of the moss to the laboratory, time and conditions of storage of the moss before extraction, part of the moss used to make the extraction, etc. There is no information about the number of replicates in the different analyses performed. References to the methods used?

Methodological information pointed out in your comment was presented in subsections 2.5, 2.8. Thanks to your comment similar additional details were provided in Materials and Methods section (Please see Line 107-110; Line 122-124 and Line 149).

Data for the background moss??

As written in the manuscript (Line 105-106), the background moss was collected from a small unpolluted stream, and the relative rbcL expression in the moss-bags was calculated relative to the control rbcL expression. Thus, Figure 2 showed a comparison of the changes in the relative rbcL expression between the three standing water dams.

Speculative. No data are given about the heavy metal content of the moss, nor about other environmental factors existing in the three reservoirs studied and that may be responsible for the differences observed between them.

We understand the reviewer’s suggestion and the arisen question is very important. As was pointed out above, data for analyzed pollutants were provided (Please see Line 116-119). The title of the Figure 2 was revised.

Where are the error bars?

Error bars were given.

How many replicates?

The number of replicates was written in subsection 2.8: “Fifty measurements were made for each characteristic per station…”.

So the SCCP and the other contaminants that could be in Zhrebchevo reservoir doesn´t affect the rbcL expresion? It would be interesant to have data from background moss.

Thank you for this valuable question. In fact, a suggestion that SCCPs seemed not to affect the rbcL expresion was incorporated in the draft of the manuscript. Finally, we decided to exclude it and not to speculate before further confirmation.

As was answered above, the relative rbcL expression in the moss-bags was calculated relative to the control rbcL expression; Figure 2 showed a comparison of the changes in the relative rbcL expression between the three standing water dams.

Yours sincerely,

Gana Gecheva and the co-authors

Round 2

Reviewer 1 Report

Specific comments:

Line 85-86: This sentence seems so lonely, as it not relevant with above and below paragraph. Better to rewrite and combine to the next paragraph, so be logically put forward the aquatic biomonitor, moss Fontinalis antipyretica. Moreover, the subsequent paragraph “ An exposition … (line 91-94)” shouldn’t be split up.

Line 92-93: GPX should be after peroxidases.

Line 107: replace w.w. with the full name.

Line 11: the temperature of the coolers should be given

Line 118: Acronyms of PBDE, SCCP is the first presence, full name should be given .

Line 186-187: If abnormal distribution of the dataset, how would you deal with it? Name the specific levels of significance.

Line 233: mm2 should be mm2

Author Response

Dear Reviewer 1,

We would like to thank you again for your patience and valuable and constructive comments and suggestions, which definitely helped to improve our manuscript!

We are now sending the second revised version of the manuscript following the reviews of Reviewer 1 and Reviewer 2. All corrections were highlighted by "Track Changes". Below is a response to the issues raised, followed by the corresponding line number and description of changes. Please see the attached file.

Reviewer 2 Report

The authors modified the article following the recommendations submitted. As a result, most of the problems it presented were solved.

The introduction has been revised, although some further changes should be made. Thus, p. ex. the state of the art with respect to micromorphological markers should be indicated, both in terms of plants in general and aquatic mosses (or F. antipyretica) in particular.

The full name of most acronyms has been provided. But some are still missing. So, it is suggested that they be reviewed to ensure that none are missing.

As a suggestion: in the Material and Methods section, the different subsections referring to the extraction and analysis of each marker can be grouped under a common heading.

Other more specific comments are included in the article pdf.

Author Response

Dear Reviewer 2,

We would like to thank you again for your patience and valuable and constructive comments and suggestions, which definitely helped to improve our manuscript!

We are now sending the second revised version of the manuscript following the reviews of Reviewer 1 and Reviewer 2. All corrections were highlighted by "Track Changes". Below is a response to the issues raised, followed by the corresponding line number and description of changes. Please see the attached file.
